# Identification of the GDP-L-Galactose Phosphorylase Gene as a Candidate for the Regulation of Ascorbic Acid Content in Fruits of *Capsicum annuum* L.

**DOI:** 10.3390/ijms24087529

**Published:** 2023-04-19

**Authors:** Yixin Wang, Zheng Wang, Sansheng Geng, Heshan Du, Bin Chen, Liang Sun, Guoyun Wang, Meihong Sha, Tingting Dong, Xiaofen Zhang, Qian Wang

**Affiliations:** 1Department of Vegetable Science, College of Horticulture, China Agricultural University, Beijing 100193, China; yixin_wang@cau.edu.cn (Y.W.); liang_sun@cau.edu.cn (L.S.); 2Beijing Key Laboratory of Vegetable Germplasms Improvement, Key Laboratory of Biology and Genetics Improvement of Horticultural Crops (North China), National Engineering Research Center for Vegetables, State Key Laboratory of Vegetable Biobreeding, Beijing Vegetable Research Center, Beijing Academy of Agriculture and Forestry Sciences, Beijing 100097, China; wangzheng@nercv.org (Z.W.); gengsansheng@nercv.org (S.G.); duheshan@nercv.org (H.D.); chenbin@nercv.org (B.C.); yuanyiwanggy@yeah.net (G.W.); shamhn@163.com (M.S.); tingtingdong315@163.com (T.D.)

**Keywords:** *Capsicum*, ascorbic acid, transcriptome, WGCNA, GGP, VIGS, CCS

## Abstract

Ascorbic acid (AsA) is an antioxidant with significant functions in both plants and animals. Despite its importance, there has been limited research on the molecular basis of AsA production in the fruits of *Capsicum annuum* L. In this study, we used Illumina transcriptome sequencing (RNA-seq) technology to explore the candidate genes involved in AsA biosynthesis in *Capsicum annuum* L. A total of 8272 differentially expressed genes (DEGs) were identified by the comparative transcriptome analysis. Weighted gene co-expression network analysis identified two co-expressed modules related to the AsA content (purple and light-cyan modules), and eight interested DEGs related to AsA biosynthesis were selected according to gene annotations in the purple and light-cyan modules. Moreover, we found that the gene GDP-L-galactose phosphorylase (GGP) was related to AsA content, and silencing *GGP* led to a reduction in the AsA content in fruit. These results demonstrated that *GGP* is an important gene controlling AsA biosynthesis in the fruit of *Capsicum annuum* L. In addition, we developed capsanthin/capsorubin synthase as the reporter gene for visual analysis of gene function in mature fruit, enabling us to accurately select silenced tissues and analyze the results of silencing. The findings of this study provide the theoretical basis for future research to elucidate AsA biosynthesis in *Capsicum annuum* L.

## 1. Introduction

Ascorbic acid (AsA), known as vitamin C, is one of the most prevalent antioxidants present in plants [1,2,3,4], which people acquire from their diet. To obtain this vitamin, humans must consume fruits and vegetables [5]. Pepper (*Capsicum annuum* L.) is widely recognized as a valuable source of antioxidants, with high concentrations of AsA [6]. As an important antioxidant in plants, AsA is involved in many physiological processes, including photosynthesis, growth, and senescence. For example, AsA functions as an electron donor in the photosynthetic electron transport chain, while also playing a protective role against reactive oxygen species and photoinhibition during photosynthesis [6,7,8].

The levels of ascorbic acid in plants are regulated by various processes, including biosynthesis, oxidative degradation, and recycling [9]. The four main AsA biosynthesis-associated pathways have been characterized in higher plants: the D-mannose/L-galactose pathway, D-galacturonic acid pathway, inositol pathway, and gulose pathway [10,11,12,13]. The AsA-biosynthesis-related pathways have been relatively well characterized, with key genes encoding enzymes catalyzing pathway reactions having been identified and cloned in different species, including *Arabidopsis thaliana* and rice [10,12,14,15].

Wheeler et al. were the first to reveal the D-mannose/L-galactose biosynthesis pathways in pea and *A. thaliana*, and clone the genes encoding the pathway enzymes [10]. The genes encoding GDP-mannose 3′,5′-epimerase (GME), which catalyzes two epimerization reactions, have been cloned in *A. thaliana* and rice [10,12,16]. Another important enzyme in the D-mannose/L-galactose biosynthesis pathway is GDP-L-galactose phosphorylase (GGP), which usually catalyzes the conversion of GDP-L-galactose to L-galactose 1-phosphate [14]. Other studies have confirmed that L-galactono-1,4-lactone dehydrogenase (GalLDH) catalyzes the final step of the AsA biosynthesis pathway in plants [17]. In a previous study, silencing of *SlGalLDH* led to a reduction in plant growth rate, but the total ascorbic acid content remained unchanged [18].

The D-galacturonic acid pathway related to AsA biosynthesis has been characterized in *A. thaliana* cell suspensions [19]. A gene encoding galacturonate reductase (GalUR), which is an NADPH-dependent enzyme, was isolated from strawberry (*Fragaria* × *ananassa*) [11]. Subsequent studies revealed that the overexpression of *GalUR* in potato and tomato led to the accumulation of AsA [20,21]. Myoinositol oxygenase 4 (MIOX4) was the first enzyme in the myoinositol pathway whose gene was cloned [13]. *MIOX4*, which encodes an enzyme that converts inositol to D-glucuronic acid, is overexpressed in *A. thaliana* cell lines, resulting in increased AsA contents [22].

GME catalyzes reactions that lead to the production of gulose in the gulose pathway that contributes to AsA biosynthesis. Moreover, L-gulono-1,4-lactone dehydrogenase activity was detected in potato tubers [12]. After AsA is synthesized in plants, it can be oxidized into monodehydroascorbic acid (MDHA) through reactions mediated by ascorbate oxidase (AO) and ascorbate peroxidase (APX). Afterwards, MDHA is either reconverted to AsA via monodehydroascorbate reductase (MDHAR) or converted to dehydroascorbic acid (DHA) via nonenzymatic reactions [9]. Furthermore, dehydroascorbic acid reductase (DHAR) can produce AsA [23]. MDHAR and DHAR both modulate AsA levels in ascorbate-deficient mutant plants [24].

The AsA accumulation is regulated through a complex process that includes biosynthesis, recycling, degradation, and transcriptional control [3,25]. The L-galactose pathway has been confirmed to be the main pathway for AsA biosynthesis in pepper fruits and leaves, which is consistent with the findings from tomato and *A. thaliana* research [3,6,25,26]. Although GalLDH has been suggested to be crucial for AsA biosynthesis in pepper, its importance has yet to be confirmed [17]. Recent studies on the AsA–GSH cycle, which mainly focused on stress resistance, have determined that this cycle can increase pepper tolerance to chilling, lead (Pb), and salinity stresses [27,28].

Although some key genes encoding regulators of AsA biosynthesis in plants have been identified, research on pepper has been very limited. In this study, transcriptome was employed for inbred line Z5 with a lower AsA content and inbred line Z6 with a higher AsA content. By analyzing the differentially expressed genes (DEGs) related to AsA synthesis, it was found that the promoter and coding sequences (CDS) of GDP-L-galactose phosphorylase (GGP) were different between Z5 and Z6. Further, *GGP* silencing reduced AsA content in fruit, indicating that *GGP* gene was the candidate gene for regulating AsA content in fruit. The purpose of this study was to identify the key candidate genes involved in pepper-fruit-mediated AsA biosynthesis. The findings reported here may provide the basis for genetic improvement and breeding of pepper plants.

## 2. Results

### 2.1. Analysis of the AsA Content

The AsA content was measured by HPLC in pepper fruit peels on days 10, 20, 30, 40, 50, and 80 after flowering. Although the level of AsA in Z6 pepper lines was significantly higher than that in Z5 pepper lines throughout the whole growth period, the pattern of AsA content accumulation in Z5 and Z6 was different. The level of AsA in the Z6 pepper strain increased throughout the growth period, while it did not change significantly in Z5 from day 50 to day 80. On day 80, the average AsA content in Z5 and Z6 was 45.84 and 240.82 mg/100 g, respectively (Figure 1 and Appendix A). The AsA levels in Z5 and Z6 fruits were lowest on day 10, the accumulation pattern of AsA content differed on day 50, and the AsA levels diverged the most on day 80. As a result, the fruits collected at these three time-points were selected for RNA-seq analysis.

### 2.2. Transcription Analysis of Z5 and Z6 at Three Time Points

In total, 18 cDNA libraries were constructed and sequenced. The raw sequencing data generated in this study are available in the NCBI Sequence Read Archive (PRJNA846120). After filtering the raw data, we obtained 60.77–76.13 million clean reads for each sample. The Q20 and Q30 values exceeded 99.09% and 96.52%, respectively, showing the high throughput and quality of the RNA-seq data. The average GC content of the reads ranged from 43.34% to 44.46% for all libraries. Additionally, 52.65–72.30 million reads were mapped to the *C. annuum* cv. Criollo de Morelos 334 reference genome (Appendix A).

DEGs were identified in Z6/Z5 based on the number of mapped reads. The DEG expression levels were provided in Appendix A. Overall, gene expression levels were compared between lines Z5 and Z6 on days 10, 50, and 80 (Figure 2). As the fruits ripened, the number of DEGs initially increased and then decreased. Ultimately, we identified 8272 DEGs on the basis of the comparative transcriptome analysis, of which, 2560 DEGs (1336 and 1224 with significantly increased and decreased expression levels, respectively) were detected on day 10 (Figure 2A), 6753 DEGs (3241 and 3512 with significantly increased and decreased expression levels, respectively) were detected on day 50 (Figure 2B), and 2688 DEGs (1314 and 1374 with significantly increased and decreased expression levels, respectively) were detected on day 80 (Figure 2C). Furthermore, the analysis of all three time-points detected 896 overlapping DEGs (Figure 2D).

### 2.3. Functional Annotation of DEGs

All DEGs were used as queries to screen the Clusters of Orthologous Groups (COG) database (http://www.ncbi.nlm.nih.gov/COG, accessed on 21 November 2019). Overall, 8272 genes were clustered into 22 functional categories (Figure 3A). After excluding the genes with unknown functions, the three largest functional categories were “post-translational modification, protein turnover, chaperones” (O, 600 genes), “transcription” (K, 557 genes), and “signal transduction mechanisms” (T, 556 genes).

The GO terms were used to classify the DEGs detected on days 10, 50, and 80 into functional categories. We annotated 2560, 6753, and 2688 genes with one or more GO terms from the biological process, cellular component, and molecular function categories, respectively (Figure 3B). Interestingly, the percentage of genes assigned each term was similar at each time point. In the biological process category, the three predominant GO terms were “metabolic processes” (30.94%, 35.76%, and 32.55% of the DEGs on days 10, 50, and 80, respectively), “cellular processes” (24.77%, 31.65%, and 28.87% of the DEGs on days 10, 50, and 80, respectively), and “single-organism processes” (19.65%, 22.61%, and 20.98% of the DEGs on days 10, 50, and 80, respectively). In the cellular component category, the most common GO terms were “cell” (21.05%, 29.34%, and 26.12% of the DEGs on days 10, 50, and 80, respectively), “cell part” (20.94%, 29.2%, and 25.97% of the DEGs on days 10, 50, and 80, respectively), and “membrane” (18.01%, 21.38%, and 19.79% of the DEGs on days 10, 50, and 80, respectively). In the molecular function category, “catalytic activity” (29.18%, 32.13%, and 29.99% of the DEGs on days 10, 50, and 80, respectively), “binding” (24.88%, 29.11%, and 27.16% of the DEGs on days 10, 50, and 80, respectively), and “transporter activity” (3.48%, 4.09%, and 3.79% of the DEGs on days 10, 50, and 80, respectively) were the main GO terms.

### 2.4. Enriched KEGG Pathways among the DEGs between Lines Z5 and Z6

To explore the main pathways mediating AsA biosynthesis in pepper, all DEGs detected on days 10, 50, and 80 were screened against the KEGG database. The predominant KEGG pathway associated with the DEGs was “metabolism” (1227 DEGs). The “carbohydrate metabolism” pathway had the most genes, which represented the upper modules of the “ascorbate and aldarate metabolism” pathway (Figure 4A).

To explore the biological pathways and functional networks involved, enrichment analyses of the KEGG pathway were performed. On day 10, 116 KEGG pathways were enriched among the 635 DEGs for the Z5 vs. Z6 comparison, with four significantly enriched pathways. The main KEGG pathways were as follows: “Phenylpropanoid biosynthesis” (37 DEGs), “Monoterpenoid biosynthesis” (6 DEGs), and “Fatty acid degradation” (17 DEGs) (Figure 4B). On day 50, 124 KEGG pathways were enriched among the 1678 DEGs for the Z5 vs. Z6 comparison, with 16 significantly enriched pathways. The predominant KEGG pathways were as follows: “Phenylalanine metabolism” (35 DEGs), “Biosynthesis of unsaturated fatty acids” (25 DEGs), and “Glycolysis/Gluconeogenesis” (58 DEGs) (Figure 4C). On day 80, 117 KEGG pathways were enriched among the 662 DEGs for the Z5 vs. Z6 comparison, including two significantly enriched pathways. The main KEGG pathways were as follows: “Pantothenate and CoA biosynthesis” (12 DEGs), “Carotenoid biosynthesis” (14 DEGs), and “Fatty acid degradation” (14 DEGs) (Figure 4D). The KEGG analysis indicated that “Fatty acid degradation”, “Phenylalanine metabolism”, and “Phenylalanine, tyrosine and tryptophan biosynthesis” were the highly enriched pathways among the DEGs between Z5 and Z6 on days 10, 50, and 80.

### 2.5. Co-Expression Network Analysis and Hub Genes Identification

The WGCNA was performed to find hub genes in modules that were associated with AsA levels. According to the dendrogram (Figure 5A), the main branches included 21 different modules, represented by different colors. Spearman correlation coefficient analyses revealed that the 21 modules were associated with different samples, with two co-expressed modules highly correlated (r ≥ 0.8) with the AsA content. More specifically, AsA was associated with the purple module (0.806) and the light-cyan module (−0.806) (Figure 5B). There were 284 and 137 genes in the purple and light-cyan modules, respectively. According to the gene annotations, four interested DEGs were selected from the purple module, including a transcription factor (AP2/ERF), an F-box gene, a calcium-binding protein CML44, and an ascorbate oxidase. Four interested DEGs were selected from the light-cyan module, including two transcription factors (bHLH) and two F-box genes (Table 1).

### 2.6. The Analysis of DEGs Related to AsA Biosynthesis Showed That GGP Was the Key Gene Affecting AsA Content

To further clarify the influence of structural genes on AsA biosynthesis, we visualized the transcriptional abundance of 65 genes associated with AsA biosynthesis. Among these genes, 19 genes were differentially expressed and the TPM value was ≥ 2 in at least one time point in Z5 or in Z6 (Figure 6 and Appendix A). These 16 DEGs encoded one GDP-mannose pyrophosphorylase (GMP), one GDP-L-galactose phosphorylase (GGP), one L-galactono-1,4-lactone dehydrogenase (GalLDH), one L-gulonolactone oxidase (GulLO), seven pectin methylesterase (PME), one D-galacturonic acid reductase (GalUR), one aldonolactonase (Alase), one L-myo-inositol 1-phosphate synthase (IPS), and two myo-inositol oxygenase (MIOX).

The expression of *GMP* and *GGP* in the L-galactose pathway was higher in Z6 than that in Z5, while the expression of GalLDH in Z5 was higher than that in Z6. In the gulose pathway, the expression of *GulLO* was higher in Z6 than that in Z5 on three time points. In the D-galacturonic acid pathway, the expression of *PME*, *GalUR* and *Alase* were higher in Z5 than in Z6; on the contrary, in the inositol pathway, the expression of *IPS* and *MIOX* were higher in Z6 than in Z5.

Interestingly, the key enzyme GDP-L-galactose phosphorylase (GGP, *CA.PGAv.1.6.scaffold65.175*) of the AsA key synthesis pathway (L-galactose pathway) was highly expressed in Z6. Furthermore, on the 10th and 50th days, *GGP* expression in Z6 was 2.42 times and 6.14 times higher than that in Z5, respectively, indicating that GGP is the key enzyme of pepper AsA synthesis. The phylogenetic analysis of GGP proteins in *Capsicum annuum* and other species showed that GGP proteins in *Capsicum annuum* had high homology with those in *Solanum melongena*, *Solanum lycopersicum*, and *Solanum tuberosum*, with a homology of 94.04%, 93.35%, 93.14%, respectively (Figure 7). GGP promoter and CDS sanger sequencing showed that Z6 had the same sequence as the reference genome. However, there were eight SNP variations in CDS between Z5 and the reference genome. These included three synonymous mutations located at +279 bp, +1038 bp and +1146 bp. Moreover, five non-synonymous mutations at +744 bp, +774 bp, +785 bp, +790 bp, and +1217 bp resulted in changes in the 248th, 258th, 262th, 264th, 406th amino acid residues, from phenylalanine (TTC) to leucine (TTA), from isoleucine (ATA) to methionine (ATG), from glycine (GGT) to aspartic acid (GAT), from arginine (AGA) to glycine (GGA), and from proline (CCT) to glycine (CGT), respectively (Appendix A). The Z5 promoter was only 92.94% similar to the genome sequence, and it had a large degree of mutation (Appendix A) affecting the expression of *GGP*. Therefore, *GGP* was the candidate gene for causing the difference in AsA content between Z5 and Z6.

### 2.7. Silencing of GGP Reduced the Content of AsA in Fruit

According to the transcriptome results, we silenced *GGP* in a high AsA pepper variety. There was no significant difference in AsA content among pTRV1 + pTRV2-CCS-R (red pericarp of plants infiltrated with pTRV1 + pTRV2-CCS), pTRV1 + pTRV2-CCS-Y (yellow pericarp of plants infiltrated with pTRV1 + pTRV2-CCS-GGP), and pTRV1 + pTRV2-R (pericarp of plants infiltrated with pTRV1 + pTRV2). The AsA content in pTRV1 + pTRV2-CCS-GGP-R (red pericarp of plants infiltrated with pTRV1 + pTRV2-CCS-GGP) was 60.83% of that in pTRV1 + pTRV2-R, and the AsA content in pTRV1 + pTRV2-CCS-GGP-Y (yellow pericarp of plants infiltrated with pTRV1 + pTRV2-CCS) was 28.68% of that in pTRV1 + pTRV2-R (Figure 8A–D). Therefore, the expression of *GGP* significantly affects the content of AsA in pepper fruit, indicating that it is the key gene for the synthesis of AsA in pepper fruit.

The expression of *CCS* in pTRV1 + pTRV2-CCS-R and pTRV1 + pTRV2-CCS-GGP-R were not significantly different from that in the negative control pTRV1 + pTRV2-R (Figure 8E). The expression of *CCS* in pTRV1 + pTRV2-CCS-R was decreased to 14.95% of that of pTRV1 + pTRV2-R, the expression of *CCS* in pTRV1 + pTRV2-CCS-GGP-Y decreased to 6.47% of that of pTRV1 + pTRV2-R (Figure 8E). The expression of *GGP* was not significantly different between pTRV1 + pTRV2-R and pTRV1 + pTRV2-CCS-R, pTRV1 + pTRV2-CCS-Y, and pTRV1 + pTRV2-CCS-GGP-R. The expression of *GGP* in pTRV1 + pTRV2-CCS-GGP-Y decreased to 17.38% of that of pTRV1 + pTRV2-R (Figure 8E).

### 2.8. Validation by qRT-PCR

A qRT-PCR investigation of 10 genes confirmed the RNA-seq results. The qRT-PCR study of these 10 genes indicated significant expression-level changes that were largely consistent with the RNA-seq results (Figure 9). Between the RNA-seq and qRT-PCR data, the mean correlation value was 0.842 (0.715–0.933) and data are presented in Appendix A. The expression pattern determined by qRT-PCR was consistent with RNA-seq (Figure 9), indicating that the data of RNA-seq were valid.

## 3. Discussion

### 3.1. Different Pepper Varieties Have Different Accumulation Patterns of AsA Content

The AsA content of pepper fruits differs significantly among varieties [29]. We measured the AsA content of pepper fruits in Z5 and Z6. The AsA content in the fruit of Z6 continued to increase throughout the development stage, but the AsA content in the fruit of Z5 remained constant at 46.48 mg/100 g after day 50 (Figure 1). The content of AsA in the pepper fruit increased during the early stage, but showed different accumulation patterns in the later stage. Our results were consistent with previous reports where, in some varieties, the AsA content continued to increase during the later stage of fruit development [3,25]. Furthermore, some maintained a constant level during the later stage [3,26], and the AsA content even decreased in some during the later stage of fruit development [30].

### 3.2. AsA Biosynthesis-Related DEGs in Pepper Lines Z5 and Z6

The DEGs related to AsA synthesis biosynthesis were studied, and only 16/63 genes were differentially expressed between Z5 and Z6. There are four AsA biosynthesis-related pathways in plants, of which the L-galactose pathway requires glucose as the starting substrate to generate AsA via reactions catalyzed by nine enzymes [10,11]. In our study, *GMP* was highly expressed on day 10 in Z6 compared with that in Z5, and the expression of *GalLDH* was decreased in Z6 compared with that in Z5. Notably, *CA.PGAv.1.6.scaffold65.175* (GGP) was stably expressed in Z5, whereas its expression level increased in Z6, peaking on day 50. Moreover, the expression of *GGP* was higher in Z6 compared with that in Z5 (Figure 6). However, it has been reported that the initial steps in the gulose pathway are the same as those in the L-galactose pathway [12]. Thus, the expression of *GulLO* was higher in Z6 than that in Z5. The D-galacturonic acid pathway uses D-glucose-6-P as the initial substrate, which is converted to L-galactono-1,4-lactone in reactions catalyzed by PG and GalUR [11]. In fully ripened fruits, pectin begins to degrade to produce UDP-D-galacturonic acid. The expression of *PME*, *GalUR*, and *Alase* genes in Z5 was higher than that in Z6 (Figure 6). The inositol pathway requires pectin as the starting substrate, which is ultimately converted to L-galactono-1,4-lactone by IPS and MIOX [13]. Our results showed that the genes encoding these enzymes were highly expressed in Z6 compared with those in Z5 (Figure 6). Therefore, the L-galactose, gulose, and inositol pathways play an important role in the accumulation of AsA content in Z6.

According to the WGCNA results, the purple and light-cyan modules were most related to the AsA content (Figure 5B). The structural gene of AsA cycling, the ascorbate oxidase gene, was found in the purple module [9]. Furthermore, it is well known that transcription factors have important functions in many crucial metabolic pathways [31]. Similarly, three transcription factors were found in the purple and light-cyan modules in our study. Among these, ethylene-responsive element binding factors can bind to the *VTC1* (*GGP*) promoter to modulate AsA biosynthesis by directly regulating the expression of the AsA-biosynthesis-related gene [32]. Recent studies have determined that *AcERF91* is highly co-expressed with *AcGGP3* in kiwi fruit [33]. The *AcERF91* TF binds to the *AcGGP3* promoter to activate gene expression. In addition, the transient expression of AcERF91 significantly increases the AsA content and *AcGGP3* expression in kiwi fruit [33]. Through genome-wide association analysis, an ascorbate quantitative trait locus *TFA9* co-localized with *SlbHLH59*, and *SlbHLH59* has been shown to directly bind to the *SlPMM*, *SlGMP2* and *SlGMP3* promoter to affect AsA content [34]. In addition, there are some proteins or enzymes that regulate the synthesis or circulation of AsA. In the ozone-sensitive *A. thaliana* mutant, an F-box protein (AMR1) reportedly decreases the AsA content by negatively regulating the expression of genes in the D-mannose/L-galactose pathway, although the underlying mechanism remains unclear [35]. However, recent studies have shown that *MdAMR1L1* interacts with *MdGMP1* and promotes its degradation via the ubiquitination pathway to inhibit AsA synthesis at the post-translational level [36]. Ca^2+^ plays an important role in plant cell structure and physiological function, and calmodulin (CaM) is an important Ca^2+^- binding sensor relay protein. Further, the yeast two-hybrid assay also found that CaM-like protein (CML10) can interact with PMM to affect AsA content [37].

### 3.3. GGP Was the Candidate Gene That Affected the AsA Content between Z5 and Z6

The L-galactose pathway is believed to play a major role in AsA metabolism in *A. thaliana*, *Ziziphus jujuba,* and tomato [38,39,40]. Transcriptome analysis showed that the expression of *GMP* in Z6 was higher than that of Z5 on day 10. However, the TPM value of *GMP*, the differentially expressed gene, was lower than that of *GMP*, the non-differentially expressed gene (Appendix A). GMP is one of the non-committed steps of the L-galactose pathway, and it has also been shown to be involved in the synthesis of cell wall polysaccharides and glycoproteins, which may reduce their impact on AsA levels [41]. The expression of *GGP* in Z6 was higher than that in Z5 on days 10 and 50 (Figure 6). Furthermore, the TPM value of *GGP* (*CA.PGAv.1.6.scaffold65.175*), the differentially expressed gene, was higher than that of *GGP*, which was not differentially expressed. This indicates that *GGP* is the gene that mainly affects the synthesis of AsA in the L-galactose pathway in Z5 and Z6 (Appendix A). The gene encoding GGP was discovered as the last identified gene in the L-galactose pathway, and it is also considered to be the key gene regulating AsA biosynthesis in plants [42,43]. A previous study has confirmed the importance of GGP for AsA accumulation in *A. thaliana* by demonstrating that the AsA level was 1- to 2-times greater in the wild-type control than in AsA-deficient vtc2 mutants [44]. In a recent study, the ascorbate content of two SlGGP1-deficient EMS Micro-Tom tomato mutants was less than half of that in the wild-type control [45]. In tomato, strawberry, potato, and tobacco, the overexpression of *GGP* increases the AsA content [42,43]. Similarly, AsA levels are correlated with *GGP* gene transcription levels in celery [46], orange [47], and apple [48]. GGP has also been shown to be the first committed step in ascorbate biosynthesis, which further proves that GGP regulates ascorbic acid biosynthesis [49]. Moreover, we found that the promoter and exon of *CA.PGAv.1.6.scaffold65.175* were different between Z5 and Z6. In particular, the promoter of Z5 had significant variation, which was only 92.94% similar to Z6. The presence of different promoters may influence the expression of downstream genes (Appendix A). The silence of *GGP* resulted in a lower AsA content in the fruit (Figure 8). As a result, *GGP* was a strong candidate gene responsible for the difference in AsA content between Z5 and Z6 fruit.

### 3.4. A Reporter for the Visual Analysis of Gene Function in Mature Fruit Based on CCS

VIGS is an easy and rapid way to study gene functions, including lethal genes, and VIGS do not need stable plant transformation [50]. The disadvantages of employing VIGS are that the phenotype of VIGS cannot be stably inherited and that VIGS is uneven or localized, leading to a lack of silencing in specific tissues [50]. The silencing efficiency of VIGS was different in different fruits of a plant, or different parts of a fruit (Figure 8C). Heterogenous coloration was also found in the CaMET1-like1-silenced and the CaPDS-silenced fruits [51]. AsA content is an invisible characteristic, and could be transported in the plant [52]. The accurate selection of the VIGS silenced region could accurately reflect the result of the silencing. As a result, *CCS* was used as a reporter to isolate silenced and non-silenced tissues to improve the accuracy of downstream analysis. The presence of capsanthin and capsorubin pigments makes mature peppers turn red. When CCS is deleted or mutated, the pigments capsanthin and capsorubin cannot be produced, causing the mature fruit to be yellow [53]. There was no significant change in AsA content and *GGP* expression in pTRV1 + pTRV2-CCS-Y and pTRV1 + pTRV2-R, while the expression of *CCS* decreased (Figure 8E). This shows that the silencing of *CCS* did not affect AsA content, and *CCS* could be used as a reporter gene for the invisible character of VIGS.

## 4. Materials and Methods

### 4.1. Plant Materials and Fruit Treatments

This study was completed using two inbred lines (*Capsicum annuum* L.), namely Z5, which has a relatively low fruit AsA content, and Z6, which has a higher fruit AsA content. Z5 and Z6 were produced by the Beijing Academy of Agricultural and Forestry Sciences. The plants for both lines were grown in the vegetable greenhouse at the Sijiqing Experimental Field (Beijing, China) in 2018, with flowers identified with tags. We harvested pepper peels of Z5 and Z6 on days 10, 20, 30, 40, 50, and 80 after the flower buds opened. At each time point, the tissues from 3 to 6 fruits were thoroughly mixed, with three biological replicates were obtained per time point. A total of 48 pepper peel samples were prepared for subsequent analysis of AsA levels. The fruit tissues were immediately frozen in liquid nitrogen and stored at −80 °C prior to the RNA extraction step.

### 4.2. Quantitative Analysis of the AsA Content

More specifically, the AsA content was measured using a high-performance liquid chromatography (HPLC) system (Shimadzu, LC-20AT, Kyoto, Japan) as previously described by Yan et al. [54], with minor modifications. The detector of the HPLC instrumentation was an ultraviolet–visible detector (SPD-20A, Kyoto, Japan). The sample solutions were passed through a 0.45 μm water-phase filter membrane and then separated on a Diamonsil C18 column (DiKMA, Beijing, China). The absorbance of a series of AsA standard working solutions (g/mL) was determined using the HPLC system to establish a standard curve. The absorbance of each sample solution was measured and used to determine the AsA concentration according to the standard curve.

### 4.3. Transcriptome Sequencing Analysis

The RNAprep Pure Kit (For Plant) (Tiangen, Beijing, China) was used to extract total RNA from 18 Z5 and Z6 pepper peel samples (three biological replicates for the fruits collected on days 10, 50, and 80; these time points were selected on the basis of the AsA contents). The purity of the isolated RNA was tested using a NanoDrop 2000 spectrophotometer (Thermo, Beijing, China), while the RNA quality and integrity were assessed using the 2100 PicoChip and Bioanalyzer (Agilent, Palo Alto, CA, USA). Oligo-(dT) beads were used to concentrate the high-quality RNA for each sample, after which, the mRNA was fragmented and cDNA was obtained via a reverse transcription reaction involving arbitrary primers and the RT enzyme (Takara, Dalian, China). The resulting cDNA was purified using the QIAquick PCR Purification Kit (Qiagen, Beijing, China) and then ligated to Illumina sequencing adapters (San Diego, CA, USA) after an end-repair step. The ligation products were separated by agarose gel electrophoresis, selected according to size, and then amplified by PCR to construct the cDNA library, which was sequenced on the Illumina HiSeq 4000™ system (BGI, Shenzhen, China).

### 4.4. Transcriptomic Analyses

The low-quality sequencing reads (Q ≤ 20 or ambiguous nucleotides), adapter sequences, and contaminating sequences were removed. The clean reads were mapped to the *Capsicum* reference genome (*C. annuum* cv. Criollo de Morelos 334) using the default parameters of HISAT (version 2.1.0) (https://ccb.jhu.edu/software/hisat2/index.shtml, accessed on 2 December 2018) [55,56]. New transcripts were assembled using the default parameters of StringTie (version 1.3.3b) (https://wiki.gacrc.uga.edu/wiki/StringTie-Sapelo, accessed on 2 December 2018) [57]. Transcript abundances were determined by the transcripts per million reads (TPM) values [58,59]. The differentially expressed genes (DEGs) were identified in Z6/Z5 at three time points, and DEGs were identified based on the number of mapped reads under the thresholds of p-adjust ≤ 0.01 and absolute log2 fold change ≥ 2 using the DESeq2 R package (https://www.bioconductor.org/packages/release/bioc/html/DESeq2.html, accessed on 21 November 2019) [60]. The p-adjust was obtained by correcting the *p*-value using the Bonferroni method. A weighted gene co-expression network analysis (WGCNA) was performed to identify hub genes in modules correlated with AsA levels [61]. Transcriptome analysis was conducted on MajorbioCloud. A total of 14,377 genes were included in the WGCNA. The structural genes of the AsA synthesis and circulation pathway were identified in the pepper genome database by BLAST, and their homologues in *Arabidopsis* were used as queries.

### 4.5. Phylogenetic Analysis

The amino acid sequences of different species of GGP were obtained from the NCBI database. Protein sequence alignment was performed via ClustalW software (version 2.1) and a neighbor-joining tree was constructed using MEGA 7 software (version 7.0.26).

### 4.6. Virus-Induced Gene Silencing

Virus-induced gene silencing (VIGS) used pTRV1 and pTRV2 vectors modified by the tobacco rattle virus (TRV) [62]. Fragments of capsanthin/capsorubin synthase (*CCS*) (250) and *GGP* (250) were amplified from cDNA and cloned into the pTRV2 vector to construct pTRV2-CCS and pTRV2-CCS-GGP plasmids. Primers were designed using Primer 5 software (Appendix A). The VIGS experiment was carried out according to Cheng et al. [63]. According to Xiao’s method, pTRV1 + pTRV2, pTRV1 + pTRV2-PDS, pTRV1 + pTRV2-CCS, and pTRV1 + pTRV2-CCS-GGP were inoculated into the cotyledons of 1-week-old pepper seedlings. The content of AsA in the pericarp was determined in mature fruit. The content of ASA in the yellow and red pericarp of plants infiltrated with pTRV1 + pTRV2-CCS and pTRV1 + pTRV2-CCS-GGP were measured, respectively.

### 4.7. Quantitative qRT-PCR Validation

The cDNA synthesized using the PrimeScript RT reagent Kit (Takara) served as the template for qRT-PCR analyses. The selected qRT-PCR primers (Appendix A), which were designed using Primer 5, produced amplicons that were 100–300 bp long. The pepper *UBI-3* gene was selected as the internal control. Relative target gene expression was analyzed using the 2^−ΔΔCt^ method [64].

### 4.8. Statistical Analysis

The data for each group were analyzed using the SPSS 22.0 software. The independent samples *t*-test was used to evaluate the significance of any differences between the mean values of the two lines. A correlation analysis was performed to determine the degree of association between the variables. The threshold for statistical significance was *p* < 0.05. All experiments were performed at least three times.

## 5. Conclusions

In this study, we conducted a transcriptome sequencing analysis of the fruits collected from two pepper lines at three time points. A total of 8272 DEGs were identified. On the basis of the WGCNA results combined with the annotations, we identified eight interested DEGs that may be related to the accumulation of AsA. It was observed that the L-galactose pathway, gulose pathway, and inositol pathway mediated the accumulation of AsA in pepper fruit. Through transcriptome, sanger-sequencing, and VIGS, *GGP* was identified as the candidate gene for regulating AsA.

## Figures and Tables

**Figure 1 ijms-24-07529-f001:**
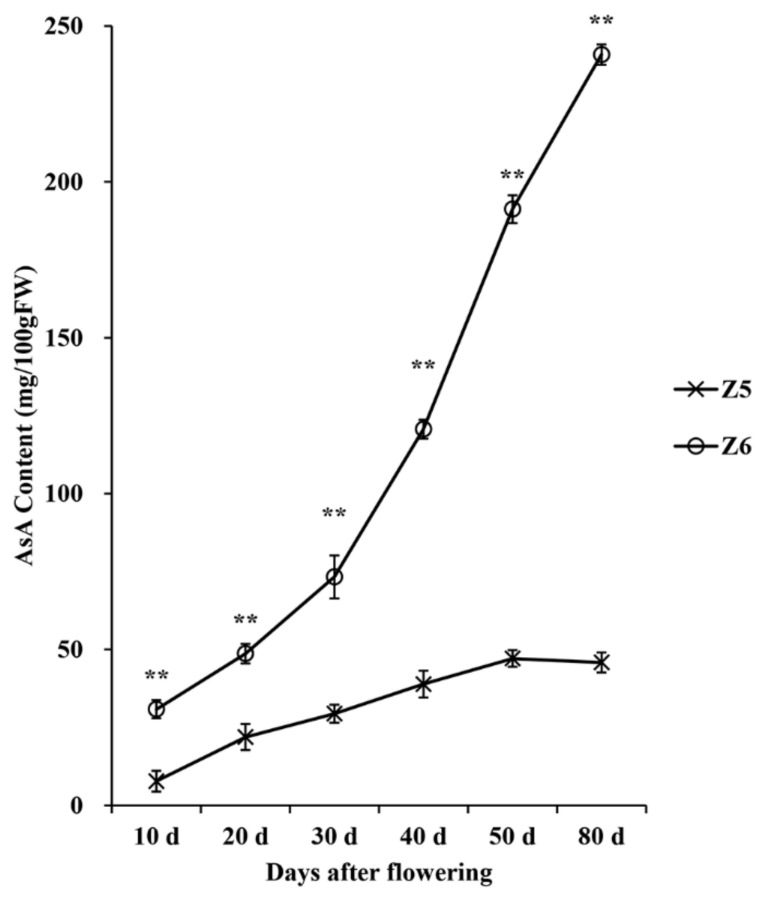
AsA concentration (mg/100 gFW) of Z5 and Z6 at different fruit development times. “**” indicates a significant difference in AsA content between Z5 and Z6 at the 0.01 level.

**Figure 2 ijms-24-07529-f002:**
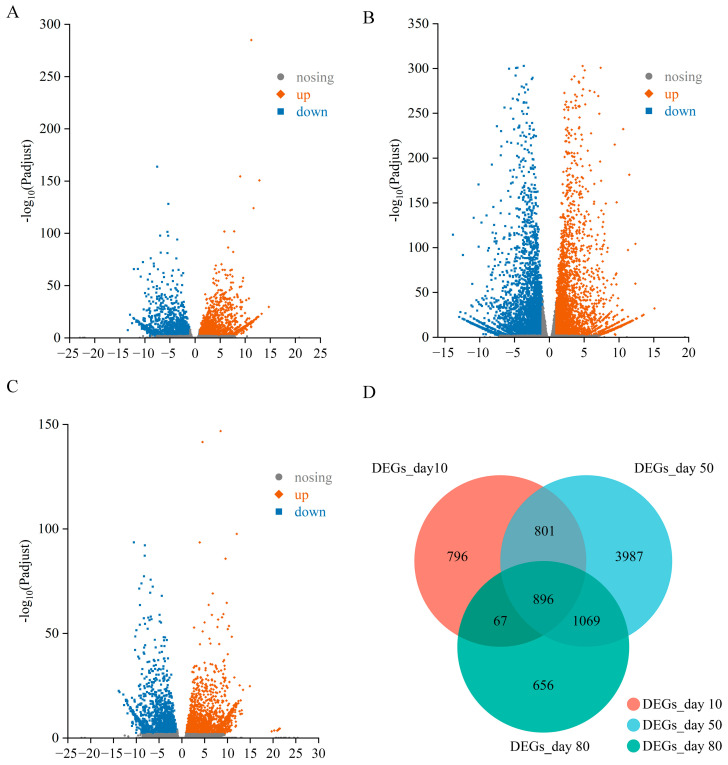
DEGs between Z5 and Z6 lines. (**A**) DEGs between Z5 and Z6 lines on day 10. (**B**) DEGs between Z5 and Z6 lines on day 50. (**C**) DEGs between Z5 and Z6 lines on day 80. (**D**) Overlap of DEGs at different times.

**Figure 3 ijms-24-07529-f003:**
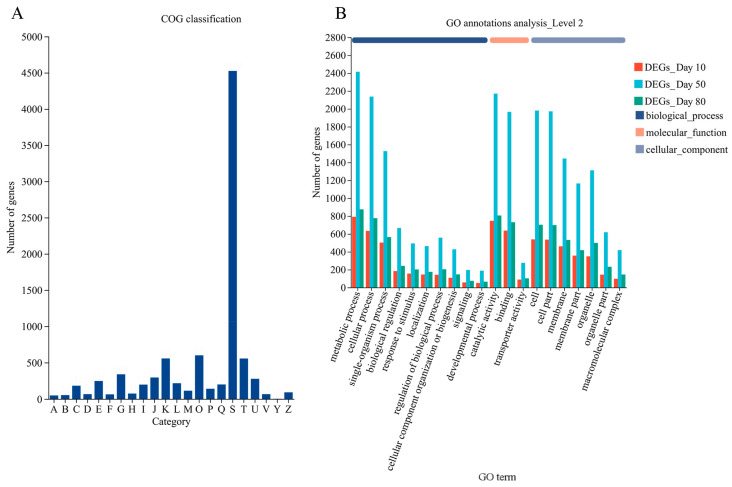
Functional annotation of DEGs. (**A**) COG classification of all DEGs (sum of DEGs between Z5 and Z6 on days 10, 50, and 80). A: RNA processing and modification; B: Chromatin structure and dynamics; C: Energy production and conversion; D: Cell cycle control, cell division, chromosome partitioning; E: Amino acid transport and metabolism; F: Nucleotide transport and metabolism; G: Carbohydrate transport and metabolism; H: Coenzyme transport and metabolism; I: Lipid transport and metabolism; J: Translation, ribosomal structure and biogenesis; K: Transcription; L: Replication, recombination and repair; M: Cell wall/membrane/envelope biogenesis; O: Post-translational modification, protein turnover, chaperones; P: Inorganic ion transport and metabolism; Q: Secondary metabolites biosynthesis, transport and catabolism; S: Function unknown; T: Signal transduction mechanisms; U: Intracellular trafficking, secretion, and vesicular transport; V: Defense mechanisms; Y: Nuclear structure; Z: Cytoskeleton. (**B**) GO annotations analyses of all DEGs.

**Figure 4 ijms-24-07529-f004:**
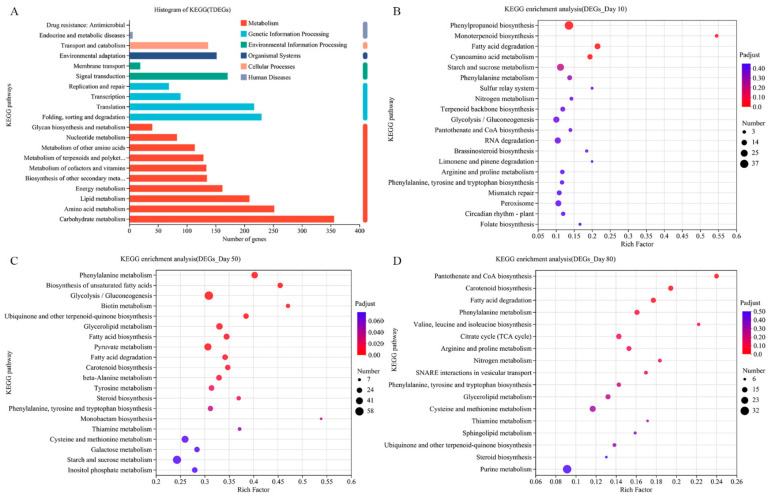
KEGG pathway enrichment analyses between Z5 and Z6 lines. (**A**) KEGG annotations analyses of all DEGs. (**B**) KEGG pathway enrichment analyses between Z5 and Z6 lines on day 10. (**C**) KEGG pathway enrichment analyses between Z5 and Z6 lines on day 50. (**D**) KEGG pathway enrichment analyses between Z5 and Z6 lines on day 80.

**Figure 5 ijms-24-07529-f005:**
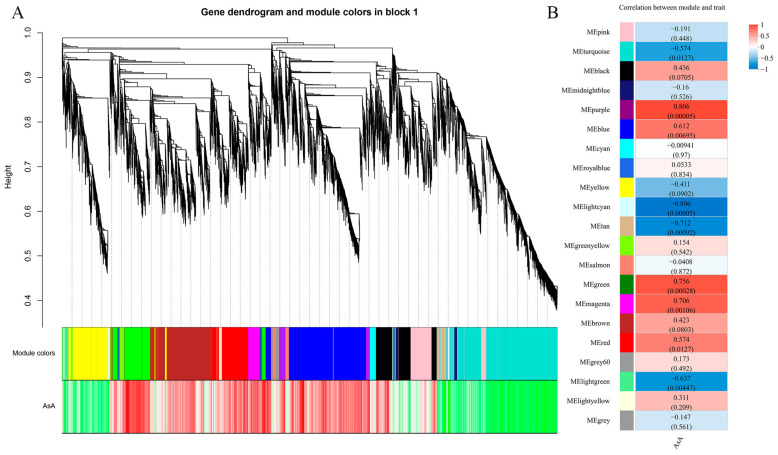
Co-expression network analysis. (**A**) A hierarchical cluster tree showing 21 modules obtained by weighted gene co-expression network analysis (WGCNA). Gray modules represent genes not divided into specific modules. Each branch in the tree points to a gene. (**B**) Module–metabolite association matrix. Gene expression profile data of AsA levels at different time points, for WGCNA analysis. Correlation coefficients and *p*-values between modules and metabolites are shown at the row–column intersection.

**Figure 6 ijms-24-07529-f006:**
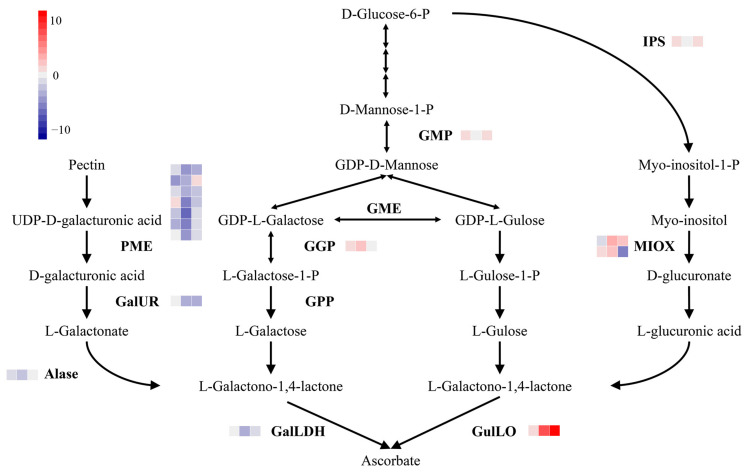
AsA biosynthetic pathway DEGs in Z5 and Z6 transcriptomes. A heat map was generated from the log2 fold change calculated from transcriptome. Each line represents a gene. The arrow direction indicates the conversion of metabolite into another. The change of color from red to gray to blue represents the log2 value from large to small. The first, second, and third columns represent log2FC(Z6/Z5) on days 10, 50, and 80. The enzymes involved in AsA biosynthesis are: GDP-mannose pyrophosphorylase (GMP), GDP-mannose-3′,5′-epimerase (GME), GDP-L-galactose phosphorylase (GGP), L-galactose-1-phosphate phosphatase (GPP), L-galactono-1,4-lactone dehydrogenase (GalLDH), L-gulonolactone oxidase (GulLO), L-myo-inositol 1-phosphate synthase (IPS), myo-inositol oxygenase (MIOX), pectin methylesterase (PME), D-galacturonic acid reductase (GalUR), and aldonolactonase (Alase).

**Figure 7 ijms-24-07529-f007:**
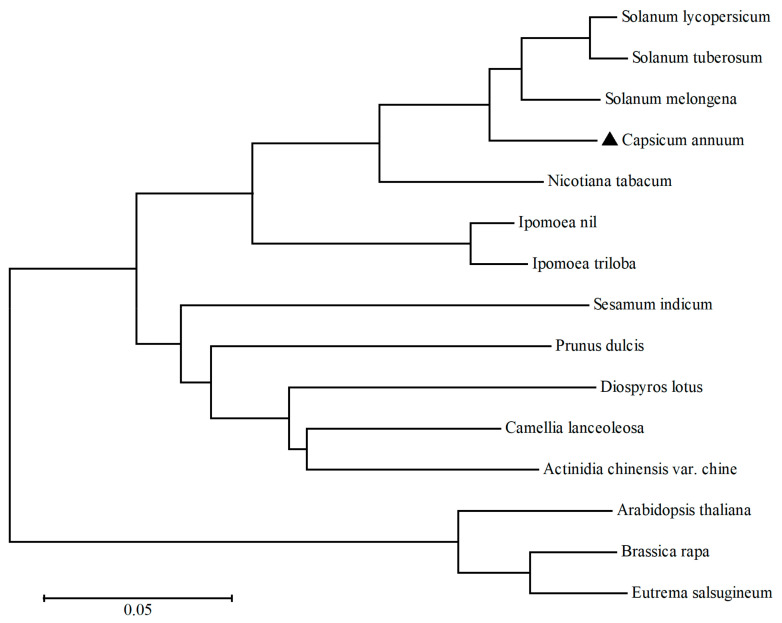
Phylogenetic tree of GDP-L-galactose phosphorylase from *Capsicum annuum* and other species. Accession number for the respective protein sequences were as follows: *Solanum melongena* (AOW42607.1); *Solanum lycopersicum* (NP_001266145.1); *Solanum tuberosum* (NP_001275300.1); *Nicotiana tabacum* (NP_001312827.1); *Ipomoea nil* (XP_019166189.1); *Ipomoea triloba* (XP_031126808.1); *Diospyros lotus* (XP_052178134.1); *Sesamum indicum* (XP_011095440.1); *Camellia lanceoleosa* (KAI8008974.1); *Prunus dulcis* (XP_034211435.1); *Actinidia chinensis* var. *chinensis* (PSR85311.1); *Arabidopsis thaliana* (NP_567759.1); *Brassica rapa* (XP_009109165.1); *Eutrema salsugineum* (XP_006413133.1); and *Capsicum annuum* (CA.PGAv.1.6.scaffold65.175, highlighted by a black triangle).

**Figure 8 ijms-24-07529-f008:**
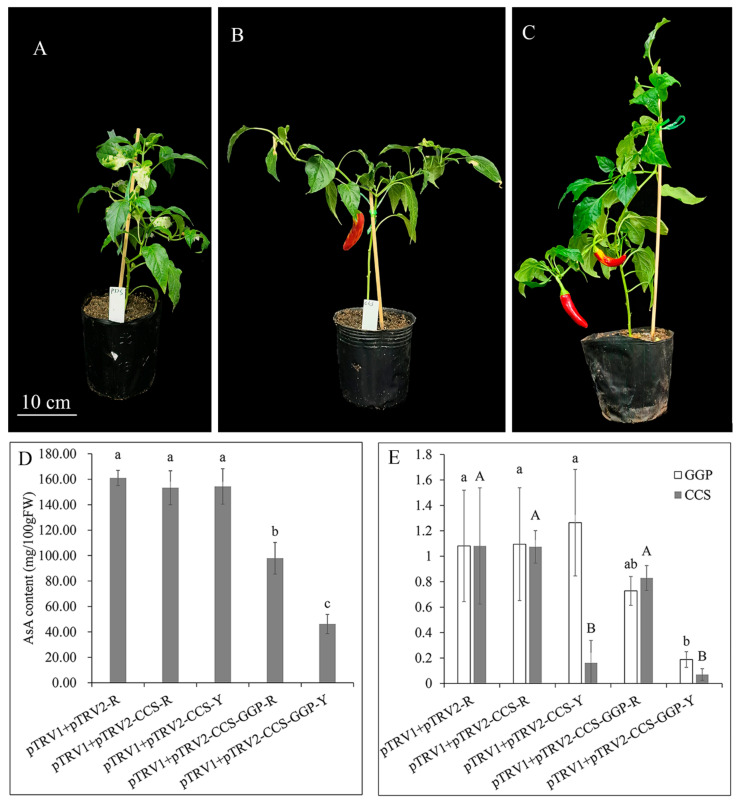
Silencing of *GGP* led to the decrease in AsA content. (**A**–**C**) Plant injected with pTRV1 + pTRV2-PDS (**A**), pTRV1 + pTRV2-CCS (**B**), and pTRV1 + pTRV2-CCS-GGP (**C**). (**D**) AsA content in pericarp of plants. Letters indicate the significance of the HSD test of the AsA content in pericarp. (**E**) Relative expression of *CCS* and *GGP* in pericarp of plants. Lowercase and capital letters indicate the significance of the HSD test in relation to the relative expression of CCS and GGP in pericarp, respectively.

**Figure 9 ijms-24-07529-f009:**
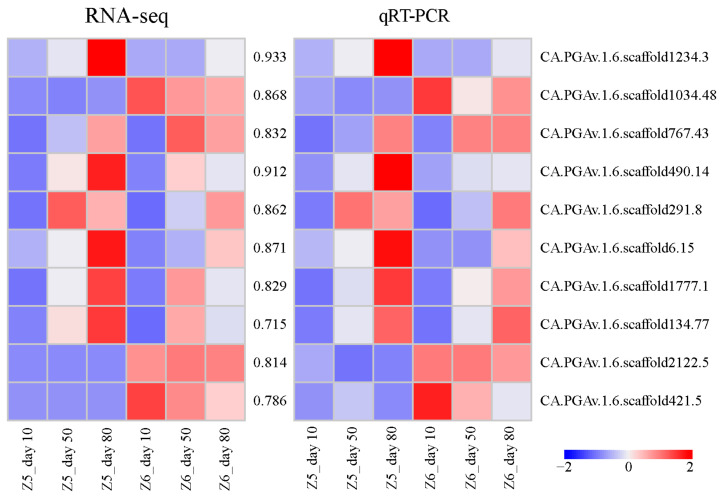
Validation of expression level of selected genes in Z5 and Z6. Heatmaps represent expression profiles of selected genes (labeled on the right side) obtained from RNA-seq (**left**) and qRT-PCR (**right**) analysis. The values between the two heatmaps represent the correlation between expression profiles of selected genes obtained from RNA-seq and qRT-PCR analysis.

**Table 1 ijms-24-07529-t001:** Interesting genes that were screen out of the purple and light-cyan module.

Gene Name	Module	Gene Description
CA.PGAv.1.6.scaffold127.3	Purple	Ethylene-responsive transcription factor ERF012
CA.PGAv.1.6.scaffold395.18	Purple	F-box/kelch-repeat protein
CA.PGAv.1.6.scaffold46.46	Purple	Probable calcium-binding protein CML44
CA.PGAv.1.6.scaffold401.2	Purple	L-ascorbate oxidase homolog
CA.PGAv.1.6.scaffold724.11	Light-cyan	Basic helix-loop-helix protein 85
CA.PGAv.1.6.scaffold843.20	Light-cyan	Basic helix-loop-helix protein 116
CA.PGAv.1.6.scaffold1305.10	Light-cyan	F-box/kelch-repeat protein SKIP11
CA.PGAv.1.6.scaffold2152.1	Light-cyan	Putative F-box protein PP2-B12

## Data Availability

The raw data of RNA-seq is available at NCBI SRA: PRJNA846120. The other raw data are available as Appendix A.

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
