# Peer review of "Identification of the GDP-L-Galactose Phosphorylase Gene as a Candidate for the Regulation of Ascorbic Acid Content in Fruits of Capsicum annuum L."

_ijms, 2023, doi:10.3390/ijms24087529_

Round 1
Reviewer 1 Report
In this study, the authors investigated the production of ascorbic acid (AsA) in pepper fruits at different developmental stages from two distinct pepper cultivars. RNA sequencing was performed to identify the key genes involved in AsA synthesis in pepper fruits. While the purpose of the study is suitable for publication, I noticed that the order of several sections in the manuscript was poorly organized, and there were numerous abbreviations that needed to be explained or spelled out. Additionally, it would be beneficial for the authors to reanalyze the DEGs using the number of mapped reads instead of TPM values, and they should refer to the DESeq2 manual for guidance. Moreover, to identify DEGs between the two pepper cultivars, an increased threshold should be used, and the results for Figures 2, 3, and 4 should be revised accordingly. I recommend this manuscript for publication after major revisions have been made. My specific comments are as follows:
Title: How about to revise the title as follows:
Identification of the GDP-L-galactose transferase gene as a candidate regulator of ascorbic acid content in fruits of Capsicum annuum L.
Abstract:
Remove "L27 (CA.PGAv.1.6.scaffold65.175)"
Spell out "GGP"
"Capsicum annuum L." should be used instead of "pepper"
L36-39:
"Ascorbic acid (AsA), known as vitamin C, is one of the most prevalent antioxidants present in plants [1-4], which people acquire from their diet."
"To obtain this vitamin, humans must consume fruits and vegetables."
L89:
Please provide more information about the choice of Z5 and Z6 pepper cultivar lines.
L90:
"DEGs" should be spelled out as "differentially expressed genes"
"CDS" should be spelled out as "coding sequences"
"GGP" should be spelled out
L98:
Please explain how the AsA level was measured and using which tissue.
L103:
Table S3 should be renamed as Table S1, and the order of all supplementary tables should be reorganized.
Table 1:
It can be moved to the supplementary tables.
L119:
The authors should identify DEGs based on the number of reads instead of TPM values using DESeq2. Please reanalyze them.
Please describe how the authors compared the conditions, for example, Z6/Z5.
Figures 2, 3, and 4:
Please reduce the number of DEGs by applying new thresholds, such as a four-fold change and adjusted p-values less than 0.01.
Create new figures based on the new results and rewrite the results section accordingly.
L225-226:
All gene names should be spelled out.
L242:
"GGP" should be spelled out.
It might be desirable to show the phylogenetic relationship of GGP proteins from other plant species.
L284-287:
The results of qRT-PCR should be presented as a figure in the main manuscript, and the results should be explained briefly.
L390-391:
This information should be included in the introduction or results.
L400-408:
This section should be presented at the beginning of the Materials and Methods section.
Please provide a detailed description of the 48 pepper peel samples, including information on the conditions and cultivars.
L434:
For DEG analysis using DESeq2, it is important to use the number of mapped reads instead of TPM values. Please reanalyze all DEGs.
Please describe how the conditions were compared for DEG analysis.
Please use log2(fold changes) to indicate the threshold for fold changes.
Reviewer 2 Report
Until recently, biochemical and, especially, pharmaceutical studies devoted to L-ascorbic acid – vitamin C – were complicated by the fact that, except humans, there is only one suitable laboratory animal that cannot synthesize its own AsA (guinea pig). A lot of metabolomic studies is done with the aim for prospective new antibiotics or other medicines and food supplements, and with the need of 5 test animal species, this area was less studied and still is quite untapped; most of current state of knowledge is only based on model plants, like A. thaliana.
From the viewpoint of foodomics, pepper fruit is an ideal source of vitamin C, rich not only with AsA, but also with compounds that support its uptake, all in the form of tasty vegetable that can be consumed raw and has a lot of use in cooking. Increasing the knowledge base about the synthesis of AsA in pepper fruit will help the producers to create ideal conditions and grow healthy functional food with balanced content of the nutrients of interest.
The paper “GGP is the candidate gene for regulating AsA content in pepper fruit in Capsicum” by Yixin Wang, Zheng Wang, Sansheng Geng, Heshan Du, Bin Chen, Liang Sun, Guoyun Wang, Meihong Sha, Tingting Dong, Xiaofen Zhang and Qian Wang describes transcriptome sequencing of two inbred lines of Capsicum annuum L., Z5 and Z6.
The content of AsA was monitored during 80-days period in two different samples, transcriptome sequencing and analysis were performed and further experiments with GGP silencing. In a comprehensive study, 12 188 DEG were identified and interesting genes related to AsA content were selected. Enrichment analyses of the KEGG pathway were performed, showing that areas important for the functional food – phenylalanine metabolism, carotenoid biosynthesis, fatty acid degradation, and glutathione metabolism – were the highly enriched pathways.
The authors proved that GGP was the key gene affecting AsA content and developed capsanthin/capsorubin synthase as the reporter gene for visual analysis.
The manuscript is given in good, simple scientific English, samples and methods are clearly described, all done according to GLP rules. The authors do not mention the type of HPLC instrumentation and detectors they have and the link to previous method shyly says “Shimadzu” with no further specification, but this is not trace analysis and no problems are expected with analytical determination. The result support the hypothesis and the data are presented clearly, often in graphical form.
This paper will be beneficial, as proposed by the authors, for the breeders who strive for vegetables with high content of vitamin C, as well as for further research of this interesting area of metabolomics, and is recommended to be accepted in the International Journal of Moleculary Science.
Comment to the authors: Add the info about your analytical instrumentation. People will not google it 3 manuscripts down just to find out. And nobody expects you to use newest OrbiTrap to measure some C-vit in rich samples.
Comment to editors: This minor revision is rather formal, just a little data will be added. :)
Author Response
Response to Reviewer 2 Comments
Until recently, biochemical and, especially, pharmaceutical studies devoted to L-ascorbic acid – vitamin C – were complicated by the fact that, except humans, there is only one suitable laboratory animal that cannot synthesize its own AsA (guinea pig). A lot of metabolomic studies is done with the aim for prospective new antibiotics or other medicines and food supplements, and with the need of 5 test animal species, this area was less studied and still is quite untapped; most of current state of knowledge is only based on model plants, like A. thaliana.
From the viewpoint of foodomics, pepper fruit is an ideal source of vitamin C, rich not only with AsA, but also with compounds that support its uptake, all in the form of tasty vegetable that can be consumed raw and has a lot of use in cooking. Increasing the knowledge base about the synthesis of AsA in pepper fruit will help the producers to create ideal conditions and grow healthy functional food with balanced content of the nutrients of interest.
The paper “GGP is the candidate gene for regulating AsA content in pepper fruit in Capsicum” by Yixin Wang, Zheng Wang, Sansheng Geng, Heshan Du, Bin Chen, Liang Sun, Guoyun Wang, Meihong Sha, Tingting Dong, Xiaofen Zhang and Qian Wang describes transcriptome sequencing of two inbred lines of Capsicum annuum L., Z5 and Z6.
The content of AsA was monitored during 80-days period in two different samples, transcriptome sequencing and analysis were performed and further experiments with GGP silencing. In a comprehensive study, 12 188 DEG were identified and interesting genes related to AsA content were selected. Enrichment analyses of the KEGG pathway were performed, showing that areas important for the functional food – phenylalanine metabolism, carotenoid biosynthesis, fatty acid degradation, and glutathione metabolism – were the highly enriched pathways.
The authors proved that GGP was the key gene affecting AsA content and developed capsanthin/capsorubin synthase as the reporter gene for visual analysis.
The manuscript is given in good, simple scientific English, samples and methods are clearly described, all done according to GLP rules. The authors do not mention the type of HPLC instrumentation and detectors they have and the link to previous method shyly says “Shimadzu” with no further specification, but this is not trace analysis and no problems are expected with analytical determination. The result support the hypothesis and the data are presented clearly, often in graphical form.
This paper will be beneficial, as proposed by the authors, for the breeders who strive for vegetables with high content of vitamin C, as well as for further research of this interesting area of metabolomics, and is recommended to be accepted in the International Journal of Moleculary Science.
Comment to the authors: Add the info about your analytical instrumentation. People will not google it 3 manuscripts down just to find out. And nobody expects you to use newest OrbiTrap to measure some C-vit in rich samples.
Comment to editors: This minor revision is rather formal, just a little data will be added. :)
Thank you very much for your approval and patient revision of this article. The Shimadzu LC-20AT HPLC system (Japan) with ultraviolet-visible detector (SPD-20A, Japan) was uesd in our study and we have added this information in the "Materials and Methods" section.
Round 2
Reviewer 1 Report
I recommend accepting the manuscript for publication as the authors have made appropriate revisions in reponse to the reviewer's comments.